# Bioassay-Guided Isolation of Iridoid Glucosides from *Stenaria nigricans*, Their Biting Deterrence against *Aedes aegypti* (Diptera: Culicidae), and Repellency Assessment against Imported Fire Ants (Hymenoptera: Formicidae)

**DOI:** 10.3390/molecules27207053

**Published:** 2022-10-19

**Authors:** Fazila Zulfiqar, Abbas Ali, Zulfiqar Ali, Ikhlas A. Khan

**Affiliations:** 1National Center for Natural Products Research, School of Pharmacy, The University of Mississippi, Oxford, MS 38677, USA; 2Department of BioMolecular Sciences, Division of Pharmacognosy, School of Pharmacy, The University of Mississippi, Oxford, MS 38677, USA

**Keywords:** mosquitoes, *Aedes aegypti*, biting deterrent, fire ant repellency, *Stenaria nigricans*, Rubiaceae, iridoid glucosides

## Abstract

In our natural product screening program, we screened natural products for their repellency and toxicity against insect vectors. Methanolic extract of aerial parts of *Stenaria nigricans* (Lam.), with no published chemistry, was tested for repellency against mosquitoes and imported hybrid fire ants. Methanolic extracts showed biting deterrence similar to DEET (*N*,*N*-diethyl-3-methylbenzamide) against *Aedes aegypti* L. Based on this activity, the crude extract was fractionated into chloroform, ethyl acetate, and methanol subfractions. The active methanolic subfraction was further fractionated into 13 subfractions. These fractions were tested for their biting deterrence against *Ae. Aegypti*. Active subfractions were further characterized to identify the compounds responsible for this activity. Four undescribed iridoid glucosides (**1**–**4**) and three previously reported compounds (**5**–**7**) were isolated from active subfractions and tested for their biting deterrent activity. Based on BDI values, compounds **2**, **3**, **6**, and **7**, with biting deterrence similar to DEET, showed the potential to be used as repellents against mosquitoes. In an in vitro digging bioassay, none of these compounds showed any repellency against hybrid imported fire ants at a dose of 125 µg/g. This is the first report of biting deterrence and repellency of *S. nigricans* extract and its pure compounds, iridoid glucosides against mosquitoes and imported fire ants. Further studies will be conducted to explore the repellent potential of these compounds in different formulations under field conditions.

## 1. Introduction

Mosquitoes are important in global public health because of their ability to transmit diseases. Mosquitoes are vectors that can cause human diseases such as malaria, dengue fever, yellow fever, and Chikungunya. In cases of high levels of transmission, epidemics can result in substantial human morbidity and mortality. In addition to other viruses, *Aedes aegypti* (L) and *Ae. albopictus* (Skuse) are vectors of the dengue and Zika viruses [1]. Dengue is one of the major vector-borne diseases that causes severe morbidity and mortality, affecting around 50–100 million people yearly [2]. Malaria, which presents a serious threat to global health, is a common disease vectored by the *Anopheles* spp. of mosquitoes [3], whereas *Culex quinquefasciatus* Say transmits the West Nile virus [4]. Imported fire ants (Hymenoptera: Formicidae) have a wide distribution in the world, and are pests of significant agricultural and medical importance. *Solenopsis invicta* Buren and *S. richteri* Forel are two fire ant species present in the United States. Extensive hybridization is reported to occur along the population boundaries between *S. invicta* and *S. richteri* in the Southern states [5,6]. The use of insect repellents such as DEET (*N*,*N*-diethyl-3-methylbenzamide), a standard among the series, significantly reduces mosquito bites, ultimately reducing the transmission of diseases [7]. A mosquito species may develop resistance to common synthetic insecticides following repeated long-term use [8]. Also, ecological aspectssuch as tropical storms, substantial rainfalls, flooding, etc. may increase mosquito breeding habitats, resulting in pest resistance alongside other health and environmental concerns [9,10,11,12,13]. Therefore, new environmentally friendly alternatives for mosquito management, particularly from natural sources, are needed. Plants are a potential source of bioactive natural compounds that possess low mammalian toxicity and are ecologically non-determined. The exploration of novel mosquito repellent from environmentally safe plant sources has been conducted at our facility [13,14,15,16,17,18]. 

Since ancient times, plants have traditionally been used for the treatment of a variety of illnesses, which has recently pushed researchers to investigate various plants and their active principles. The Rubiaceae family comprises over 13,000 species in more than 600 genera and ranks fourth in species diversity among angiosperms. Iridoids are abundant, along with anthraquinones, triterpenes, and alkaloids found in the Rubiaceae family, particularly in the *Uncaria*, *Psychotria*, *Hedyotis*, *Ophiorrhiza*, and *Morinda* genera [19]. *Stenaria* is a small genus in Rubiaceae with seven species native to Mexico, the United States, and the Bahamas [20]. *Stenaria* plants are herbaceous small shrubs/perennials with pink, purple, or white flowers, first recognized as a genus in 2001. The comprised species were formerly classified as *Hedyotis* or *Houstonia* [21].

*Stenaria nigricans* (Lam.) Terrell (syn. *Houstonia nigricans* (Lam.) Fernald, *Hedyotis nigricans* (Lam.) Fosberg), commonly known as fine leaf bluet, narrow leaf bluet, diamond flower, prairie bluet, and glade bluet, is a perennial with a woody taproot. *S. nigricans*, the main species of the genus, is prevalent in central and northern Mexico and the central, eastern, and southwestern United States. None of the *Stenaria* species have been investigated for phytochemistry or evaluated for biological potential, whereas the *Hedyotis* genus has been well-studied [22]. In our natural product screening program against mosquitoes, hundreds of in-house plant extracts were randomly screened for biting deterrent activity in Klun and Debboun (K&D) bioassays against *Aedes aegypti* L. *Stenaria nigricans* extract showed significant biting deterrent activity prompting the investigation of its chemical constituents. Seven iridoid glucosides (Figure 1), including four undescribed ones (**1**–**4**), were isolated through bio-guided fractionation of the methanol extract of *S. nigricans*. Structure elucidation was achieved by NMR and mass spectral data analysis. Iridoids are specialized metabolites that usually exist as glycosides and structurally belong to monoterpenoids containing a cyclopentanopyran moiety. A bicyclic H-5/H-9 β,β-*cis*-fused cyclopentanopyran ring system is the most common structural feature of iridoids [23]. There are a few reports where the repellency and insecticidal activity of iridoids against mosquitoes have been reported [24,25]. This study reports the systematic bio-guided isolation and characterization of seven iridoid glucosides from the methanol extract of *S. nigricans* and their biting deterrent activity against the yellow fever mosquito, *Ae. aegypti*, and repellency against hybrid imported fire ants.

## 2. Results and Discussion 

Methanolic crude extract of *S. nigricans* aerial parts at 10 µg/cm^2^ showed biting deterrence similar to DEET at 25 nmol/cm^2^ in K & D bioassay (Figure 2a). To follow the activity, the methanolic crude extract was fractionated by vacuum liquid chromatography (VLC) over silica gel with chloroform, ethyl acetate, and methanol to yield three respective subfractions (A–C), which were tested for their biting deterrent activity (Figure 2b). Fraction C, which showed biting deterrence similar to DEET (Figure 2b), was further fractionated by silica gel column chromatography (CC) into 13 fractions (C1–C13). Fraction C4 turned out to be pure compound **1**. Except for Fr. C5 (insufficient amount), the other 12 fractions (C1–C4 and C6–C13) were tested for their biting deterrence against *Ae. aegypti* females (Figure 3). Frs. C2, C3, C6, C7, C8, and C13 at 10 µg/cm^2^ showed biting deterrence activity similar to DEET (Figure 3). Subsequently, the bioactive fractions were subjected to chromatographic techniques to find the active principles that resulted in the purification of compounds **2**–**7**. The purified compounds were characterized by spectroscopic and spectrometric techniques (1D- and 2D-NMR and HRESIMS). Compounds **1**–**3** and **5**–**7** were tested for their biting deterrent activity at 25 nmol/cm^2^. Based on BDI values, compounds **2**, **3**, **6**, and **7** showed biting deterrence similar to DEET (Figure 4) and have the potential to be used as repellents against mosquitoes. In an in vitro digging bioassay, none of these compounds showed any repellency against hybrid imported fire ants at a dose of 125 µg/g. This is the first report of biting deterrence and repellency of *S. nigricans* extract and its pure compounds, iridoid glucosides, against mosquitoes and imported fire ants. Further studies should be conducted to explore the repellent potential of these compounds in different formulations under field conditions.

Stenigroside A (**1**) was obtained as an amorphous powder and its molecular formula, C_21_H_28_O_11_, was deduced from the [M+HCOOH-H]^−^ ion peak at *m*/*z* 501.1626 (calculated for C_22_H_29_O_13,_ 501.1608) in the HRESIMS, indicating an equivalence of eight double bonds. The ^13^C-NMR spectrum of **1** displayed 21 resonances, of which ten were assignable to the iridoid skeleton, five to a 2-methylbutanoyl group, and six to a sugar unit (Table 1). The ^1^H- and ^13^C-NMR data showed resonances, attributed to the iridoid skeleton, an acetal function [*δ*_H_/*δ*_C_ 6.28 (d, *J* = 1.5 Hz)/93.1 (CH-1)], two olefins [*δ*_H_/*δ*_C_ 7.61 (d, *J* = 2.2 Hz)/149.6 (CH-3) and 5.72 (br d, *J* = 1.9 Hz)/128.6 (CH-7) and *δ*_C_ 105.7 (C-4) and 143.5 (C-8)], oxymethylene [*δ*_H_/*δ*_C_ 4.66 (dd, *J* = 14.3, 1.3 Hz) and 4.77 (dd, *J* = 14.3, 1.5 Hz)/61.2 (CH_2_-10)], oxymethine [*δ*_H_/*δ*_C_ 5.46 (dt, *J* = 6.7, 1.9 Hz)/84.8 (CH-6)], two methines [*δ*_H_/*δ*_C_ 3.51 (td, *J* = 6.7, 2.2 Hz)/36.9 (CH-5) and 3.43 (dd, *J* = 6.7, 1.5 Hz)/45.0 (CH-9)], and a carbonyl [*δ*_C_ 170.3 (C-11)]. The ^1^H- and ^13^C-NMR spectra also exhibited resonances for a 2-methyl butanoyl group [*δ*_H_/*δ*_C_ 0.88 (t, *J* = 7.4 Hz)/12.1 (CH_3_-4″), 1.16 (d, *J* = 7.0 Hz)/17.0 (CH_3_-5″), 1.45 (m) and 1.71 (m)/27.2 (CH_2_-3″), 2.44 (m)/41.4 (CH-2″), and *δ*_C_ 176.1 (C-1″)] (Table 1 and Table 2). The chemical shifts and coupling constant values related to the sugar moiety were typical of the *β*-glucopyranose unit [26] (Table 1 and Table 2). The γ-lactone was supported by the HMBC correlation observed between H-6 (*δ*_H_ 5.46) and C-11 (*δ*_C_ 170.3). The placement of the sugar unit at C-1 and methyl butanoate moiety at C-10 was confirmed by HMBC correlations of the anomeric proton (*δ*_H_ 5.34) with C-1 (*δ*_C_ 93.1) and oxymethylene protons Ha/Hb-10 (*δ*_H_ 4.66 and 4.77) with carbonyl (*δ*_C_ 176.1), respectively. Based on these results and a detailed study of the NMR data, the structure of compound **1** was determined to be similar to asperuloside (**5**) [27], except the resonances of the acetyl group were absent, instead showing resonances for a 2-methylbutanoyl group in **1**. The complete assignment of ^1^H- and ^13^C-NMR spectroscopic data (Table 1 and Table 2) was done based on ^1^H-^1^H COSY couplings and HMBC correlations (Figure 5). Based on NOESY correlations (Figure 6), the relative configuration of **1** was found to be the same as that of asperuloside (**5**) [28], i.e., cis arrangement of ring junction protons (H-5, H-6, and H-9) and *β* axial orientation of glycosidic linkage. Ultimately, the structure of stenigroside A (**1**) was elucidated, as shown in Figure 1.

Stenigroside B (**2**) was isolated as an amorphous powder. Based on the [M + HCOOH-H]^−^ ion peak observed in the HRESIMS at *m*/*z* 515.1778 (calculated for C_23_H_31_O_13_, 515.1765), its molecular formula was established as C_22_H_30_O_11_, indicating eight degrees of unsaturation. The ^1^H- and ^13^C-NMR spectroscopic data of **2** (Table 1 and Table 2) were found to be similar to those of **1**, except for the missing resonances of the 2-methylbutanoyl group in **2**, instead showing resonances for a caproyl group [*δ*_H_/*δ*_C_ 0.81 (t, *J* = 7.0 Hz)/14.3 (CH_3_-6″), 1.20 (m)/22.8 (CH_2_-5″), 1.22 (m)/31.7 (CH_2_-4″), 1.64 (p, *J* = 7.4 Hz)/25.0 (CH_2_-3″), 2.39 (t, *J* = 7.4 Hz)/34.3 (CH_2_-2″), and *δ*_C_ 173.3 (C-1″)]. The HMBC correlations of oxymethylene protons (H_2_-10) with carbonyl (*δ*_C_ 173.3) (Figure 5) confirmed that a caproyl ester moiety was located at C-10. The assignment of ^1^H- and ^13^C-NMR chemical shifts of **2** was completed based on ^1^H-^1^H COSY couplings and HMBC correlations (Figure 5). Finally, the structure of stenigroside B (**2**) was elucidated, as shown in Figure 1.

Stenigroside C (**3**), isolated as an amorphous powder, showed an [M + HCOOH-H]^−^ ion peak at *m*/*z* 491.1409 (calculated for C_20_H_27_O_14,_ 491.1401) in the HRESIMS, corresponding to a molecular formula of C_19_H_26_O_12_. The ^1^H- and ^13^C-NMR spectroscopic data of **3** (Table 1 and Table 2) were identical to those of asperuloside **5** [27], except for the resonances of a double bond between C-3 and C-4, which were replaced with those of an aliphatic methine [*δ*_H_/*δ*_C_ 3.58 (dd, *J* = 10.4, 3.3 Hz)/43.7 (CH-4)], an acetal function [*δ*_H_/*δ*_C_ 5.40 (d, *J* = 3.3 Hz)/97.8 (CH-3)], and a methoxy group [*δ*_H_/*δ*_C_ 3.70 (s)/56.5 (-3-OCH_3_)]. The methoxy group at C-3 was confirmed by HMBC correlations of H-1, H-5, and methoxy protons with C-3 (Figure 6). The NOESY correlations of H-5 with H-9, H-6, and H-4 indicated a cis arrangement of all ring junction protons, as shown in Figure 6, which consecutively supported the α-orientation of the methoxy group due to the NOESY correlation observed between H-4 and H-3. Thus, the structure of stenigroside C (**3**) was determined, as shown in Figure 1.

The molecular formula of stenigroside D (**4**), C_16_H_21_NaO_11_, was determined from an [M+HCOOH-H]^−^ ion peak at *m*/*z* 457.0962 (calculated for C_17_H_22_NaO_13_, 457.0958) in the HRESIMS. The ^13^C-NMR spectrum of **4** displayed 16 resonances ascribable to a sugar unit, an acetal function (*δ*_C_ 99.4, C-1), a carbonyl (*δ*_C_ 176.0, C-11), four olefins [*δ*_C_ 116.4 (C-4), 129.8 (C-7), 147.3 (C-8), and 150.0 (C-3)], an oxymethine (*δ*_C_ 83.1, C-6), an oxymethylene (*δ*_C_ 61.5, C-10), and two aliphatic methines [*δ*_C_ 47.2 (C-9) and 48.3 C-5)]. The ^13^C-NMR data of **4** (Table 1) was found to be comparable with that of the scandoside [29], which contained a -COOH group, except for a significant deshielding of C-4 (+5.4 ppm) and C-11 (+3.8 ppm) and shielding of C-3 (−4.1 ppm) in **4** that verified the anionization of the C-4 carboxyl group. The anionization effect has been reported to cause a deshielding impact on ^13^C-NMR resonances close to the negative charge [30,31]. The stereochemistry of **4**, assigned by the NOESY experiment (Figure 6), was the same as that of scandoside. Thus, the structure of stenigroside D (**4**) was elucidated, as shown in Figure 1.

Known compounds were characterized as asperuloside (**5**) [27], deacetylasperuloside (**6**) [32,33], and daphylloside (**7**) [34] by their NMR and mass spectral data analysis, which also matched reported values.

## 3. Material and Methods

### 3.1. General Procedures

IR spectra (frequency range 4000–500 cm^−1^) were recorded on an Agilent Technologies Cary 630 FTIR. Optical rotations were carried out at 28 °C chamber temperature on an AUTOPOL II automatic polarimeter (Rudolph Research Analytical, Hackettstown, NJ, USA). UV spectroscopy was performed on a Thermo Scientific Evolution 201 UV-Vis spectrophotometer at ambient temperature. Mass data were determined on an Agilent Technologies 6230 ToF mass spectrometer. NMR spectra were recorded at 25 °C on a Bruker AU III 500 MHz NMR spectrometer using CD_3_OD or C_5_D_5_N. Chemical shifts were referenced to the residual solvent signals of methanol and pyridine. Column chromatography was performed using flash silica gel (40–63 μm, 60 Å, SiliCycle Inc., Quebec, QC, Canada) and Sephadex LH-20 (GE Healthcare, Chicago, IL, USA), with analytical grade solvents from Fisher Scientific. Thin layer chromatography (TLC) was performed on a silica gel F_254_ aluminum sheet (20 cm × 20 cm, 200 μm, 60 Å, Sorbtech, Norcross, GA, USA) or a RP-18 silica F_254_ aluminum sheet (20 cm × 20 cm, 150 μm, 60 Å, Sorbtech, Norcross, GA, USA). The spots on the silica card were visualized by spraying with 0.5% vanillin (Sigma, St. Louis, MO, USA) solution in concentrated H_2_SO_4_-EtOH (5:95), followed by heating (≈130 °C). DEET was purchased from Sigma-Aldrich (St Louis, MO, USA).

Adults of *Ae. aegypti* used in these studies were obtained from the laboratory colonies maintained at the Mosquito and Fly Research Unit at the Center for Medical, Agricultural and Veterinary Entomology, USDA-ARS, Gainesville, Florida. For biting deterrence and repellent bioassays, eggs were hatched and the larvae were reared to adults in the laboratory. They were maintained at 27 ± 2 °C and 60 ± 10% RH, with a photoperiod regimen of 12:12 h (L:D). Eight- to eighteen-day-old adult females were used in these bioassays. Hybrid fire ant workers used in these studies were obtained from the mounds located under natural field conditions at the University Field Station, University of Mississippi, 15 County Road 2078, Abbeville, Mississippi 38601.

### 3.2. Plant Material

The aerial parts of *Stenaria nigricans* (Specimen # MOBOT 270) were obtained from the Missouri Botanical Garden, Missouri, USA. A sample specimen (# 6824) was deposited in the repository at the National Center for Natural Products Research, University of Mississippi.

### 3.3. Extraction and Isolation

The plant powder (69 g) was extracted with methanol (3 L × 20 h) at room temperature, including sonication for 60 min. The crude extract (12.5 g) was obtained after removal of the solvent under reduced pressure at 45 °C, followed by lyophilization. The extract (11.9 g) was fractionated by vacuum liquid chromatography (VLC) over silica gel (30 cm × 7 cm), with chloroform (2 L), ethyl acetate (1 L), and methanol (2 L), to yield three subfractions (A–C). The active subfraction eluted with methanol (C, 10.3 g) was chromatographed on silica gel (85 cm × 6 cm), using mixtures of EtOAc-CHCl_3_-MeOH-H_2_O [10:6:4:1 (3 L), 6:4:4:1 (6 L)] then with methanol (4 L), to afford 13 fractions (C1–C13). Compound **1** (200.3 mg) was found in a pure state in fraction C4. Compound **2** (61.8 mg) was purified from fraction C3 (130.1 mg) by column chromatography (CC) over silica gel (100 cm × 2.5 cm) using a mixture of CHCl_3_-MeOH (9:1). Compounds **3** (19.0 mg), **5** (819.8 mg), **6** (160.1 mg), and **7** (45.9 mg) were obtained from fraction C6 (1.4 g) by CC [silica gel (85 cm × 3 cm), CHCl_3_-MeOH-H_2_O (8:2:0.25 and 7:3:0.5)]. Fraction C7 (400 mg) was purified by CC [Sephadex LH-20 (100 cm × 3.5 cm), MeOH], followed by CC [silica (90 cm × 1.5 cm), CHCl_3_-MeOH-H_2_O (8:2:0.25 and 7:3:0.5)] to give compound **6** (12.0 mg). Compound **4** (500.1 mg) was purified from fraction C13 (2.1 g) by CC over Sephadex LH-20 (100 cm × 3.5 cm) with a mixture of MeOH-H_2_O (1:1).

### 3.4. Spectral Data

***Stenigroside A (1)***. Amorphous powder; [*α*]^24^_D_ -156.3 (c 0.45, MeOH); UV (MeOH) *λ*_max_ (log *ε*) 234 (3.86) nm; IR *ν*_max_ 3371, 2922, 1735, 1654, 1176, 1149, 1067, 976 cm^−1^; HRESIMS *m*/*z* 501.1626 [M+HCOOH-H]^−^ (calculated for C_22_H_29_O_13,_ 501.1608); for ^1^H- and ^13^C-NMR data, see Table 1 and Table 2.

***Stenigroside B (2)***. Amorphous powder; [α]^24^_D_ -137.7 (c 0.48, MeOH); UV (MeOH) *λ*_max_ (log *ε*) 234 (3.76) nm; IR *ν*_max_ 3373, 2926, 1742, 1653, 1168, 1169, 1010, 974 cm^−1^; HRESIMS *m*/*z* 515.1778 [M+HCOOH-H]^−^ (calculated for C_23_H_31_O_13_, 515.1765); for ^1^H- and ^13^C-NMR data, see Table 1 and Table 2.

***Stenigroside C (3)***. Amorphous powder; [*α*]^24^_D_ -34.6 (c 0.31, MeOH); IR *ν*_max_ 3350, 2926, 1735, 1654, 1595, 1228, 1067, 1004 cm^−1^; HRESIMS *m*/*z* 491.1409 [M+HCOOH-H]^−^ (calcd for C_20_H_27_O_14,_ 491.1401); for ^1^H- and ^13^C-NMR data, see Table 1 and Table 2.

***Stenigroside D (4)***. Amorphous powder; [*α*]^24^_D_ -31.5 (c 0.47, MeOH); UV (MeOH) *λ*_max_ (log *ε*) 228 (3.68) nm; IR *ν*_max_ 3274, 2894, 1638, 1541, 1395, 1340, 1038 cm^−1^; HRESIMS *m*/*z* 457.0962 [M+HCOOH-H]^−^ (calculated for C_17_H_22_NaO_13,_ 457.0958); for ^1^H- and ^13^C-NMR data, see Table 1 and Table 2.

### 3.5. In Vitro K & D Biting Deterrent Bioassay

Bioassays were conducted using a six-celled in vitro Klun and Debboun (K&D) module bioassay system for quantitative evaluation of biting deterrence [35]. Briefly, the assay system consists of a six-well reservoir with each of the 3 cm × 4 cm wells containing 6 mL of feeding solution. The reservoirs were covered with a layer of collagen membrane (Devro, Sandy Run, Swansea, SC, USA). The test compounds were applied to six 4 cm × 5 cm marked areas of organdy cloth (G Street Fabrics, Rockville, MD, USA) and positioned over the collagen-covered CPDA-1+ATP solution [15]. A six-celled K&D module containing five female mosquitoes per cell was positioned over treated organdy, covering the six CPDA-1+ATP solution membrane wells, and trap doors were opened to expose the treatments to these females. The number of mosquitoes biting through cloth treatments in each cell was recorded after a 3 min exposure. The crude preparations were evaluated at dosages of 10 µg/cm^2^, and pure compounds including DEET were tested at a concentration of 25 nmol/cm^2^. Sets of 5 replications, each with 5 females per treatment, were conducted on 2–3 different days using a newly treated organdy and a new batch of females in each replication. Treatments were repeated 10 times.

**Statistical Analyses.** Proportion not biting (PNB) was calculated using the following formula:$$\mathrm{PNB}=1-{(\;\frac{\mathrm{Total}\;\mathrm{number}\;\mathrm{of}\;\mathrm{females}\;\mathrm{biting}\;}{\mathrm{Total}\;\mathrm{number}\;\mathrm{of}\;\mathrm{females}})}^{}$$

The K&D module bioassay system could only handle four treatments, along with negative and positive controls, in order to make direct comparisons among test samples and compensate for variation in overall response among replicates; biting deterrent activity was quantified as the biting deterrence index (BDI) [15]. The BDI was calculated using the following formula:$$\left[{\mathrm{BDI}}_{i,\;j,\;k}\right]=\left[\frac{{\mathrm{PNB}}_{i,\;j,\;\;k}-{\mathrm{PNB}}_{c,j,k}}{{\mathrm{PNB}}_{d,j,k}-{\mathrm{PNB}}_{c,j,k}}\right]$$
where PNB*_i,j,k_* denotes the proportion of females not biting when exposed to test compound *i* for replication *j* and day *k* (*I* = 1–4, *j* = 1–5, *k* = 1–2), PNB*_c,j,k_* denotes the proportion of females not biting the solvent control “*c*” for replication *j* and day *k* (*j* = 1–5, *k* = 1–2), and PNB*_d,j,k_* denotes the proportion of females not biting in response to DEET “*d”*(positive control) for replication *j* and day *k* (*j* = 1–5, *k* = 1–2). This formula adjusts for inter-day variation in response, and incorporates information from the solvent control as well as the positive control. BDI values not significantly different from 1 are similar to DEET. Data were analyzed using SAS Proc ANOVA [single factor: test compound (fixed)] [36]. To determine whether confidence intervals included the values of 0 or 1 for treatments, Scheffe’s multiple comparison procedure with the option of CLM was used in SAS.

### 3.6. In Vitro Digging Bioassay for Fire Ants

Ants were identified by analysis of venom alkaloid and cuticular hydrocarbon profiles, as described by Chen et al [37]. A digging bioassay was used to determine the repellency of the pure compounds against imported fire ants. The digging bioassay used in this study was described by Ali et al [38]. The sand of a uniform size of 1 mm was washed in de-ionized water and dried at 150 °C. An amount of 4 g of sand was treated in a 45 mL aluminum weighing dish (Fisher Scientific, 300 Industry Drive, Pittsburgh, PA, USA) at a volume of 400 µL. After the evaporation of ethanol, de-ionized water was added at a rate of 0.6 µL/g of sand to moisten the sand. The vials were filled with treated sand, whereas the sand in the control treatment was only treated with ethanol. The vials were then screwed to the caps attached to the bottom of the arena. Fifty hybrid fire ant workers were released in the center of the arena petri dish. The experiment was conducted at 25 ± 2 °C temperature and 50 ± 10% relative humidity. The pure compounds were screened at our standard screening dose of 125 µg/g. After 24 h, sand was collected back into aluminum dishes, dried at 150 °C for 1 h, and weighed. 

## 4. Conclusions

Methanolic extract of aerial parts of *Stenaria nigricans* and its pure compounds showed biting deterrence against *Aedes aegypti.* Seven compounds, including four undescribed iridoid glucosides (**1**–**4**), were isolated through mosquito-biting deterrent bioassay-guided isolation. High biting deterrence of stenigroside B, stenigroside C, deacetylasperuloside, and daphylloside indicated a high potential of these natural compounds to be used as mosquito repellents. Further research, through intensive laboratory and field trials in in vivo bioassays and field trials, is needed to explore the potential of these natural products against mosquitoes. However, none of these compounds showed digging suppression against fire ants.

## Figures and Tables

**Figure 1 molecules-27-07053-f001:**
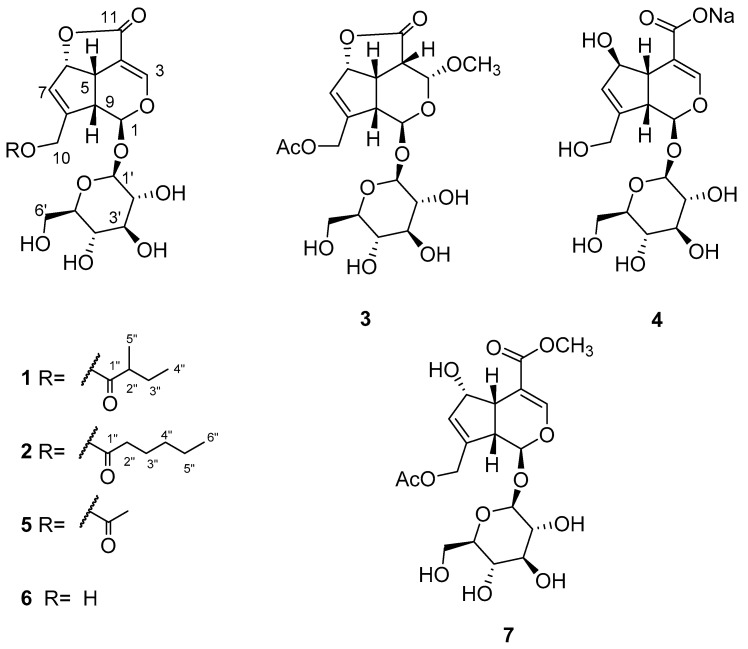
Structures of iridoids **1**–**7**.

**Figure 2 molecules-27-07053-f002:**
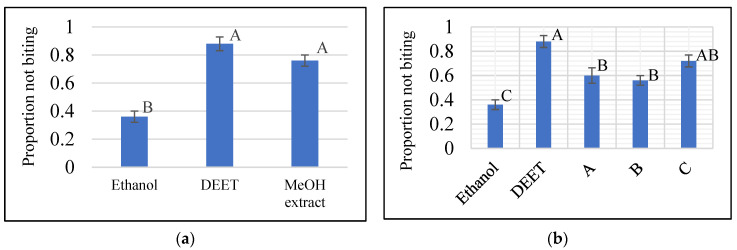
(**a**) Values of proportion not biting of methanolic extract and (**b**) subfractions of *Stenaria nigricans* and DEET against *Aedes aegypti*. The extract and subfractions were tested at 10 µg/cm^2^. DEET at 25 nmol/cm^2^ was used at a positive control and ethanol was used as a solvent control. Means within each group followed by the same letter are not significantly different (Ryan-Einot-Gabriel-Welsch multiple range test (*p* ≤ 0.05).

**Figure 3 molecules-27-07053-f003:**
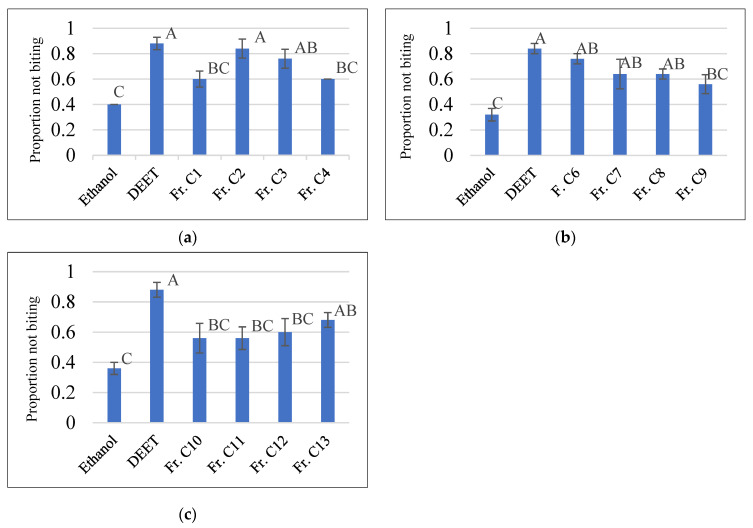
Proportion not biting values of fractions of methanolic extract part C of *Stenaria nigricans* and DEET against *Aedes aegypti*. Subfigures (**a**–**c**) represent data of fractions 1–4, 6–9 and 10–13, repectively. Ethanol was used as a solvent control. The fractions were tested at 10 µg/cm^2^. DEET at 25 nmol/cm^2^ was used as a positive control. Means within the bars in each group followed by the same letter are significantly different (Ryan-Einot-Gabriel-Welsch multiple range test (*p* ≤ 0.05).

**Figure 4 molecules-27-07053-f004:**
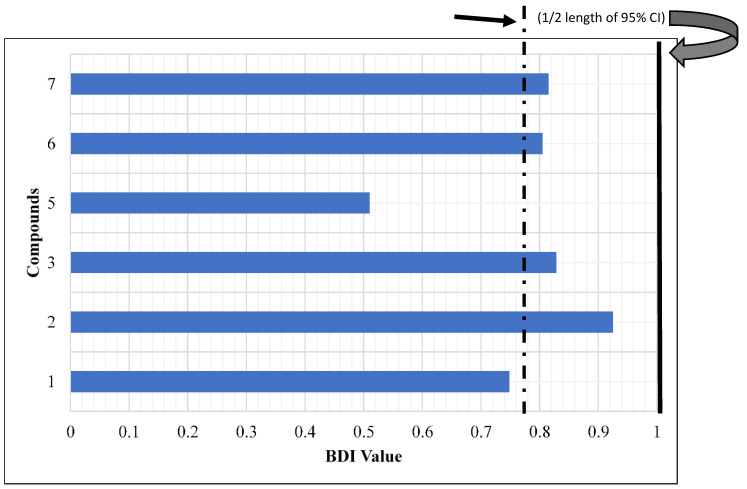
Mean biting deterrence index (BDI) values of the pure compounds purified from active fractions of methanolic extract of *Stenaria nigricans* against female *Aedes aegypti*. Ethanol was the solvent control and DEET at 25 nmol/cm^2^ was used as the positive control.

**Figure 5 molecules-27-07053-f005:**
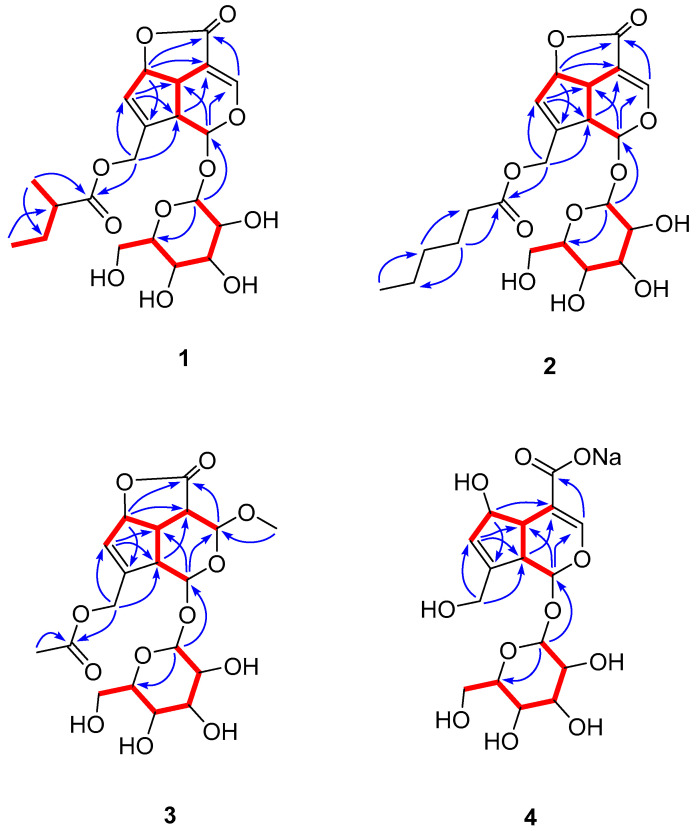
Bold bonds (▬) represent COSY couplings and curved arrows (→) indicate key HMBC correlations of compounds **1**–**4**.

**Figure 6 molecules-27-07053-f006:**
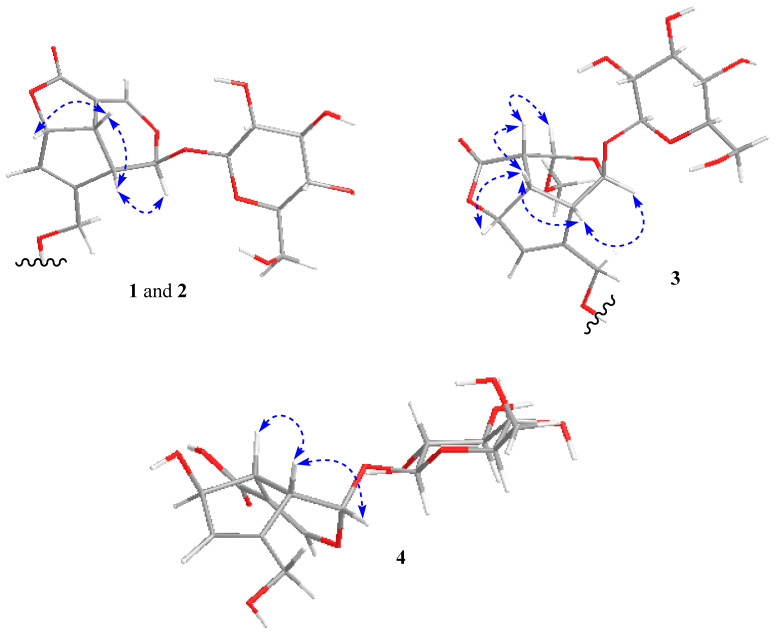
Dotted double-head arrows represent selected NOESY correlations of compounds **1**–**4**.

**Table 1 molecules-27-07053-t001:** ^13^C-NMR spectroscopic data (*δ*_C_, multiplicity) for compounds **1**–**4** (Appendix A).

Position	1 ^a^	2 ^a^	3 ^a^	4 ^b^
1	93.1, CH	93.1, CH	96.3, CH	99.4, CH
3	149.6, CH	149.5, CH	97.8, CH	150.0, CH
4	105.7, C	105.6, C	43.7, CH	116.4, C
5	36.9, CH	36.9, CH	36.8, CH	48.3, CH
6	84.8, CH	84.8, CH	86.6, CH	83.1, CH
7	128.6, CH	128.4, CH	125.4, CH	129.8, CH
8	143.5, C	143.5, C	151.4, C	147.3, C
9	45.0, CH	44.6, CH	45.6, CH	47.2, CH
10	61.2, CH_2_	61.2, CH_2_	62.0, CH_2_	61.5, CH_2_
11	170.3, C	170.3, C	175.4, C	176.0, C
1′	101.0, CH	101.0, CH	100.0, CH	100.2, CH
2′	75.1, CH	75.1, CH	75.0, CH	74.8, CH
3′	78.8, CH	78.7, CH	78.5, CH	77.6, CH
4′	71.7, CH	71.6, CH	71.5, CH	71.3, CH
5′	79.2, CH	79.2, CH	78.6, CH	78.1, CH
6′	62.9, CH_2_	62.8, CH_2_	62.7, CH_2_	62.4, CH_2_
1″	176.1, C	173.3, C	171.0, C	-
2″	41.4, CH	34.3, CH_2_	20.8, CH_3_	-
3″	27.2, CH_2_	25.0, CH_2_	-	-
4″	12.1, CH_3_	31.7, CH_2_	-	-
5″	17.0, CH_3_	22.8, CH_2_	-	-
6″	-	14.3, CH_3_	-	-
3-OCH_3_	-	-	56.5	-

^a^ Measured in C_5_D_5_N; ^b^ Measured in CD_3_OD

**Table 2 molecules-27-07053-t002:** ^1^H-NMR spectroscopic data (*δ*_H_, multiplicity (*J* in Hz)) for compounds **1**–**4**.

Position	1 ^a^	2 ^a^	3 ^a^	4 ^b^
1	6.28, d (1.5)	6.28, br s	5.52, d (5.9)	4.86, d (8.7)
3	7.61, d (2.2)	7.61, d (2.0)	5.40, d (3.3)	7.34, br s
4	-	-	3.58, dd (10.4, 3.3)	-
5	3.51, td (6.7, 2.2)	3.52, td (6.7, 2.0)	3.50, ddd (10.4, 9.2, 6.6)	2.91, dd (7.7, 6.5)
6	5.46, dt (6.7, 1.9)	5.45, br d (6.7)	5.34, br d (6.7)	4.56, br d (6.5)
7	5.72, br d (1.9)	5.73, br s	5.96, br s	5.85, br s
8	-	-	-	-
9	3.43, dd (6.7, 1.5)	3.45, br d (6.7)	3.25, dd (9.2, 5.9, 1.2)	2.81, dd (8.7, 7.7)
10	4.66, dd (14.3, 1.3) 4.77, dd (14.3, 1.5)	4.69, d (14.4) 4.79, d (14.4)	4.90, d (16.0) 5.20, d (16.0)	4.20, d (15.4) 4.39, d (15.4)
11	-	-	-	-
1′	5.34, d (7.9)	5.35, d (7.8)	5.29, d (7.9)	4.74, d (7.9)
2′	4.07, dd (8.5, 7.9)	4.09, dd (8.2, 7.8)	4.02, dd (8.7, 7.9)	3.25, dd (9.2, 7.9)
3′	4.26, dd (8.8, 8.5)	4.27, dd (8.7, 8.2)	4.22, dd (9.0, 8.7)	3.42, dd (9.2, 8.8)
4′	4.32, dd (9.3, 8.8)	4.31, dd (8.7, 8.2)	4.24, t (9.0)	3.32 ^c^
5′	4.00, ddd (9.3, 5.2, 2.5)	4.01, ddd (8.2, 5.3, 2.4)	3.87, m	3.30, m
6′	4.40, dd (11.8, 5.2) 4.52, dd (11.8, 2.5)	4.40, dd (11.8, 5.4) 4.55, dd (11.8, 2.4)	4.35, dd (11.9, 5.1) 4.50, dd (11.9, 2.5)	3.66, dd (12.0, 5.3) 3.86, dd (12.0, 2.0)
1″	-	-	-	-
2″	2.44, m	2.39, t (7.4)	-	-
3″	1.45, m1.71, m	1.64, p (7.4)	-	-
4″	0.88, t (7.4)	1.22, m	-	-
5″	1.16, d (7.0)	1.20, m	-	-
6″	-	0.81, t (7.0)	-	-
3-OCH_3_	-	-	3.70, s	-

^a^ Measured in C_5_D_5_N; ^b^ Measured in CD_3_OD; ^c^ Multiplicity not clear for some signals, due to overlapping.

## Data Availability

Not applicable.

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
