# Peer review of "Bioassay-Guided Isolation of Iridoid Glucosides from Stenaria nigricans, Their Biting Deterrence against Aedes aegypti (Diptera: Culicidae), and Repellency Assessment against Imported Fire Ants (Hymenoptera: Formicidae)"

_molecules, 2022, doi:10.3390/molecules27207053_

Round 1

Reviewer 1 Report

The method of activity tracking is effected to find active substances. In this manuscript, novel iridoids with certain activities were found from aerial parts of Stenaria nigricans. Obviously, the activity of the four compounds is significantly lower than that of the methanol extract. The reason may be that the activity of the methanol extract is the function of the mixture of these iridoids, or there are active substances that have not been identified. Here are some minor suggestions:

1.  Where are the proportion not biting values of these seven iridoids.

2. In the new iridoids, the configuration of sugars should provide more evidence. For example, compare with the standard after hydrolysis.

3. The background color in figure 6 is not suitable.

4. Whether variance data analysis can be provided in the data in figure 4.

Author Response

We appreciate your careful review comments.  Please see the response as attached file.

Reviewer 2 Report

The manuscript entitled "Bioassay-Guided Isolation of Iridoid Glucosides from Stenaria nigricans, Their Biting Deterrence Against Aedes aegypti (Diptera: Culicidae) and Repellency Assessment Against Imported Fire Ants (Hymenoptera: Formicidae)" is a good work carried out by the authors. The work described the efficacy and mechanism of biting deterrence induced by the iridoid glucosides present in S. nigricans. However, there require some clarifications and modifictions in the present mansucript. The specific comments are given below;

1. The abstract does not describe any background information; I suggest to include the background of the study in one or two lines.

2. Introduction can be started by providing a description on vector borne diseases and their impact on human health and mortality. Statistical information can be provided.

3. A more clear information must be provided about vector and pathogen in the first paragraph of the present mansucript.

4. The alignment of Figure 2 and 3 has some problems. Kindly rectify it.

5. The figure 5 and 6 are having limitted description as legends

6. The use of fire ants as a model must be mentioned. Also, details must be included in introduction.

7. The future perspectives of the study may be inlcuded in the conclusion section

8. The chemical characterization conditions must be included in detail in the methodology section

Author Response

(The authors gave the same response as above.)

Reviewer 3 Report

The submitted manuscript describes the bioactivity-guided isolation of iridoids from Stenaria nigricans. The investigated plant is a Mexican species, that has not been phytochemically recognized so far. Thus, the selection of the species is one of the novelty element of the work.

The isolated constituents belong to the class of iridoids and four of them have not been reported before. The structural novelty cannot however be assessed as high as in two cases, it results from a different acid component in previously described iridoid ester (asperuloside), and one of the new constituents is a natrium salt of already known scandoside.

Overall, the manuscript is well written. The methodology seems correct. The results are clearly presented and scientifically sound. However, in my opinion the work lacks some discussion of the results in a broader context, e.g.:

·         As there are no phytochemical studies on the genus Stenaria, it might be interesting for the Reader, if the Authors added some chemotaxonomical information, e.g. if the similar constituents are present in closely related genus Hedyotis. Moreover, as caproyl and 2-methylbutanoyl esters seem not very common, it would be interesting to know, if there are cases of such linkages elsewhere in plant kingdom.

·         The investigated extracts and some of the isolated constituents had repellent activity towards mosquitos, but not fire ants. Generally, iridoids, e.g. nepetalactone, have been previously found to have repellent properties against various insects and even some molecular mechanisms of that activity have been recognized (see e.g. https://doi.org/10.1016/j.cub.2021.02.010). A short discussion, presenting the Authors’ ideas on the possible mode of action of their constituents in the context of their properties and the activity results would add value to the paper.  

Author Response

(The authors gave the same response as above.)

Round 2

Reviewer 2 Report

No more comments as the queries were well addressed.